# Community Integrated Energy System Multi-Energy Transaction Decision Considering User Interaction

Yuantian Li and Xiaojing Wang *

School of Electrical Engineering, Xinjiang University, Urumqi 830017, China
* Correspondence: wangxiaojing345@163.com; Tel.: +86-180-9961-3605

**Abstract:** With the gradual liberalization of China's energy market, the distributed characteristics of each entity in the community integrated energy system are more and more obvious, and the traditional centralized optimization is difficult to reveal the interaction between the entities. This paper aims to improve the profit of the community operator and the users' value-added benefit of energy use, and proposes a multi-energy transaction decision of a community integrated energy system considering user interaction. First, a refined model of user interaction, including energy conversion, is established, and then the optimization model of multi-energy transaction decision between the community operator and the users is constructed based on the master–slave game. The upper layer aims to maximize the profit of the community operator according to the energy use strategies' feedback from the users, decides the retail energy prices of the community operator to the users, and optimization variables include equipment output and energy purchased from the power grid and natural gas grid. The lower layer aims to maximize the value-added benefit of energy use for users. The users optimize their energy use strategies based on the retail energy prices published by the community operator. The model is solved by the differential evolution algorithm combined with the CPLEX solver. Finally, different scenarios are analyzed in a numerical example, and the results show that the strategy proposed in this paper to set community prices increases the community operator's profit and profit margin by 5.9% and 7.5%, respectively, compared to using market energy prices directly. At the same time, the value-added benefit to users also increases by 15.2%. The community operator and users can achieve a win–win situation.

**Keywords:** integrated energy system; multi-energy trading; consumer psychology; convertible load

## 1. Introduction

With the disadvantages of low economic benefits and high energy consumption of traditional energy systems becoming increasingly prominent, integrated energy systems (IES) that can realize flexible energy conversion and efficient utilization have become the focus of energy research and development [1]. The community integrated energy system (CIES) near the user side contains a variety of energy coupling equipment, which couples and complements electricity, heat, natural gas, and other energy sources, enabling local consumption of renewable energy and providing users with comprehensive energy services. It is an important direction for the future development of the intelligent community [2]. Therefore, it has become a hot research issue to study how to improve the economics of the community integrated energy system, and to formulate transaction strategies between the community operator and the users to guide the users to rationally use energy to achieve a win–win situation [3,4].

At present, domestic and foreign scholars have focused on improving the economics of the integrated energy system, mainly on the refined modeling of equipment on the power supply side and demand side management. In terms of refined equipment modeling, this technology proposes a general dynamic energy efficiency model of an integrated energy

system, which lays the foundation for the optimization of integrated energy system operation and the formulation of trading strategies [5]. Chen et al. established an optimization model considering coupled dynamic energy efficiency, and the results showed that considering the dynamic energy efficiency of equipment can improve the energy utilization rate, which is more consistent with the actual optimization results [6]. In demand side management, Wang Yongli et al. considered the electric and heating demand response, and the established source–load interactive model can reduce the operating cost of the service provider and improve wind power consumption [7]. Guo Zihao et al. considered the multi-energy flow coupling characteristics and the users' flexible load and considered the source–load interaction to optimize the operator's benefit [8]. Liu et al. considered the controllable degree of flexible load in the scheduling process and the constraint of user satisfaction to give full play to the demand response ability of the users. The results show that the proposed scheduling scheme can reduce the cost of the community operator [9]. It can be seen that coordinated optimization on both sides of supply and demand can improve the system economy. However, the references [7–9] are mostly centralized optimization and do not consider the energy trading and pricing problems of the operator in the market environment. There is a lack of research on the impact of the operator's retail energy prices on users' energy-use strategies.

With the development of the electricity market, the community operator can be regarded as a distribution-side entity or retail entity with self-production and self-selling capabilities [10]. It can guide users to participate in interaction by formulating reasonable retail energy prices, adjusting energy use strategies, and achieving demand side management. Aiming at the transaction problem between the operator and the users, Li Yuan et al. used the master–slave game method to establish an operator energy pricing model, including electric vehicles and P2G, which can improve the system economy [11]. References [12,13] analyzed the interaction mechanism between the community operator and the users in the electricity market based on the master–slave game model, with the operator as the leader and the users as the follower. In reference [14], the master–slave game model of the transaction between the community operator and consumers was established, and transaction strategies considering the demand response ability of consumers were proposed. Fu et al. constructed a user model containing four types of loads: electricity, heat, cold, and gas. Combined with the operator revenue optimization model, a master–slave game pricing mechanism between operators and users is proposed [15]. Fleischhacker et al. proposed an energy value allocation and stabilization algorithm based on a cooperative game. By investing in distributed energy, community operators can share value among their members [16]. Based on the master–slave game, Anoh et al. constructed an energy trading strategy between operators and consumers in the microgrid to optimize the interests of producers and consumers [17]. Wei et al. proposed a multi-leader and multi-follower Stackelberg game approach to solve the multi-energy trading problem. Multiple energy operators act as leaders to determine real-time energy prices, while multiple consumers act as followers to optimize their energy usage strategies [18]. However, the above models do not consider the role of convertible load in the process of user interaction. Li Peng et al. included convertible loads in consideration of integrated demand response, which improved user interaction but did not consider the impact of energy prices on convertible load [19]. Under the incentive of multiple retail energy prices, users will preferentially use energy with lower prices to meet the same energy demand. The actual amount of interaction will be influenced by consumer psychology [20], so considering the convertible load can further tap into the potential of user interaction.

Based on the above research, this paper establishes a refined model of user interaction considering energy conversion and constructs an optimization model of multi-energy transaction decisions between the community operator and the users based on the master–slave game. Taking the community operator as the leader, the optimization goal is the maximum daily profit, and the optimization variables are retail energy prices, equipment output, etc. The users are the followers, the optimization goal is the maximum value-added

benefit of energy use, and the optimization variables are the users' energy use strategies of electricity, heat, and natural gas. Finally, the validity of the proposed model is verified by an example.

The rest of this paper is organized as follows. Section 2 summarizes the models of community integrated energy system and of user interaction. Section 3 establishes the optimization model for the multi-energy transaction decision between the community operator and the users. Section 4 sets up different scenarios to analyze the trading strategy proposed in this paper. Finally, the conclusions are drawn in Section 5.

## 2. Models of Community Integrated Energy Systems and User Interactions

### 2.1. CIES Model Based on Energy Hub

The concept of an energy hub (EH) was first proposed by Geidl et al. [21], it simplifies the energy flow relationship. An EH with multiple input and output ports is modeled by a coupling matrix that can easily describe the transformation and coupling relationship between energy input and output [22]. Therefore, to analyze the energy coupling and input–output energy flow relationship in the system, the energy hub model is used to describe the CIES model abstractly, as shown in Figure 1. In this paper, the electricity, heat, and natural gas demanded by the users in winter are supplied by the community operator in CIES, who has certain renewable energy units according to natural and geographical conditions. In the actual operation process, the community operator purchases electric energy and natural gas energy from the energy market and uses the energy conversion equipment to convert the energy into the energy required by the users according to the multi-energy complementary characteristics. The renewable energy equipment of CIES includes wind turbines (WT) and photovoltaic (PV); energy conversion equipment includes combined heat and power (CHP) units, gas boilers (GB), and electric heat pumps (EHP); energy storage devices include electricity storage and heat storage.

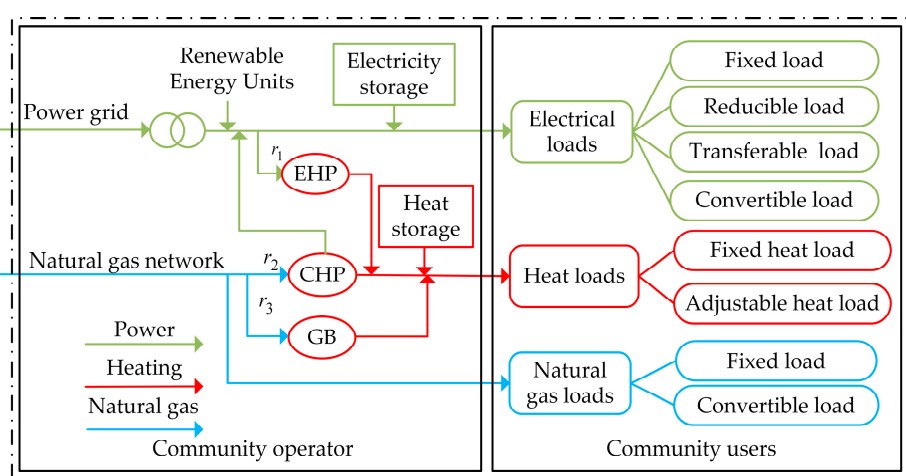

**Figure 1.** Community Integrated Energy System Model.

According to the energy flow, the community energy supply model can be represented by the following matrix:

$$\begin{bmatrix} P_{\text{out},t}^{\text{e}} \\ P_{\text{out},t}^{\text{h}} \\ P_{\text{out},t}^{\text{g}} \end{bmatrix} = \begin{bmatrix} 1-r_1 & 0 & r_2\eta_{\text{CHP,e}} \\ r_1\eta_{\text{EHP}} & 1 & r_2\eta_{\text{CHP,h}}+r_3\eta_{\text{GB}} \\ 0 & 0 & 1-r_2-r_3 \end{bmatrix} \begin{bmatrix} P_{\text{in},t}^{\text{e}} \\ P_{\text{in},t}^{\text{h}} \\ P_{\text{in},t}^{\text{g}} \end{bmatrix} - \begin{bmatrix} P_{\text{c/d,e},t}^{\text{ES,n}} \\ P_{\text{c/d,h},t}^{\text{ES,n}} \\ 0 \end{bmatrix} \tag{1}$$

In the above formula: $P_{\text{out},t}^{\text{e}}$, $P_{\text{out},t}^{\text{h}}$, and $P_{\text{out},t}^{\text{g}}$ are the electricity, heat, and natural gas power supplied by the community to the users, respectively; $P_{\text{in},t}^{\text{e}}$ is the sum of purchased power $P_{\text{net},t}^{\text{e}}$, wind power $P_{\text{w},t}^{\text{e}}$, and photovoltaic power $P_{\text{v},t}^{\text{e}}$; $P_{\text{in},t}^{\text{h}}$ is the heat power purchased by the community. This paper considers that the heat energy is only supplied

by the community, so it is taken as 0; $P_{\text{in},t}^{\text{g}}$ is the power of the community to purchase natural gas; $P_{\text{c/d,e},t}^{\text{ES,n}}$ and $P_{\text{c/d,h},t}^{\text{ES,n}}$ are the actual charging and discharging power of electric and thermal energy storage respectively; $r_1$, $r_2$, and $r_3$ are dispatch factors, which represent the proportion of heterogeneous energy flow into energy conversion equipment; $\eta_{\text{EHP}}$ and $\eta_{GB}$ are the efficiency of electric heat pump and gas boiler; $\eta_{\text{CHP,e}}$ and $\eta_{\text{CHP,h}}$ are the electrical and thermal efficiencies of the CHP unit.

*2.2. The Models of User Interaction That Account for Energy Conversion*

User interaction in energy communities makes sense and can improve the economy of the community operator [8,11]. In this paper, the community operator guides users to adjust energy use strategies by formulating reasonable retail energy prices. In general, user interaction strategies are often limited to responses in a single form of energy, such as load reduction and transfer, which have a greater impact on users' actual energy use. With the development of user terminal equipment, the user side can realize the conversion of energy forms. When the community operator publishes retail energy prices, the multi-energy complementary users consider the energy conversion efficiency of the terminal equipment to obtain the difference in equivalent energy prices. Users can use the corresponding terminal equipment to achieve secondary energy conversion and choose the appropriate way to meet their own load demand. For example, users can choose to use electric heating or natural gas heating to achieve the same hot water demand according to the equivalent electricity price and the equivalent natural gas price, etc. At the same time point, the users' actual energy use demands do not change, and the users' actual energy use has little impact, which can improve the flexibility and economy of the users' energy use. Therefore, it is of great significance to construct user-interactive models that consider the conversion of energy use.

According to the above analysis, multi-energy complementary users can choose to convert, reduce, transfer, and other ways to achieve interaction.

2.2.1. Convertible Load Model

This paper considers the users' electricity–gas convertible load and improves the convertible model of reference [23]. That is, the influence of consumer psychology is considered when optimizing the convertible load model. Based on the principle of consumer psychology, the difference between the equivalent electricity price and the equivalent natural gas price affects the response of the user's convertible load. The users' interactive response range is divided into saturation zone, linear zone, and dead zone [24,25]. When the difference between the equivalent electricity price and the equivalent natural gas price is lower than the dead zone threshold, users are unwilling to respond interactively. When it exceeds the threshold, users start to respond interactively. In the linear zone, the users' convertible load increases with the difference between the equivalent electricity price and the equivalent natural gas price, and it shows a linear upward trend. When the compensation limit is exceeded, the users' electricity–gas convertible amount tends to be saturated. In this paper, the energy use rate $\lambda_{\text{con},t}^{\text{g,e}}$ of the users' gas load to electric load and the energy use rate $\lambda_{\text{con},t}^{\text{e,g}}$ of the users' electric load to gas load are used to characterize the influence of the difference between the equivalent electricity price and the equivalent natural gas price $\pi_{\text{con},t}$ on the users' mode of energy use.

$$\lambda_{\text{con},t}^{\text{g,e}} = \begin{cases} 1 & \pi_{\text{con},t} \leq -\pi_{\text{con,max}}^{\text{g,e}} \\ \frac{\pi_{\text{con},t} + \pi_{\text{con,min}}^{\text{g,e}}}{\pi_{\text{con,min}}^{\text{g,e}} - \pi_{\text{con,max}}^{\text{g,e}}} & -\pi_{\text{con,max}}^{\text{g,e}} < \pi_{\text{con},t} < -\pi_{\text{con,min}}^{\text{g,e}} \\ 0 & -\pi_{\text{con,min}}^{\text{g,e}} \leq \pi_{\text{con},t} \leq 0 \end{cases} \qquad (2)$$

$$\lambda^{e,g}_{con,t} = \begin{cases} 0 & 0 \leq \pi_{con,t} \leq \pi^{e,g}_{con,min} \\ \dfrac{\pi_{con,t} - \pi^{e,g}_{con,min}}{\pi^{e,g}_{con,max} - \pi^{e,g}_{con,min}} & \pi^{e,g}_{con,min} < \pi_{con,t} < \pi^{e,g}_{con,max} \\ 1 & \pi^{e,g}_{con,max} \leq \pi_{con,t} \end{cases} \tag{3}$$

In the above formula: $\pi^{g,e}_{con,min}$, $\pi^{e,g}_{con,min}$, $\pi^{g,e}_{con,max}$, and $\pi^{e,g}_{con,max}$ are the dead zone threshold and saturation zone limit of the difference between the equivalent electricity price and the equivalent natural gas price when the users respond to convertible load.

According to the calorific value equivalence theorem and the energy conservation theorem, the constraints that the convertible load needs to satisfy are as follows:

$$\begin{cases} L^e_{con,t} = L^{e,n}_{con,t} - \mu^{e,g}_{con,t}\Delta L^e_{con,t} + \mu^{g,e}_{con,t}\Delta L^{g,e}_{con,t} \\ L^g_{con,t} = L^{g,n}_{con,t} - \mu^{g,e}_{con,t}\Delta L^g_{con,t} + \mu^{e,g}_{con,t}\Delta L^{e,g}_{con,t} \\ \Delta L^e_{con,t} = \lambda^{e,g}_{con,t}L^{e,n}_{con,t} \\ \Delta L^g_{con,t} = \lambda^{g,e}_{con,t}L^{g,n}_{con,t} \\ \Delta L^{g,e}_{con,t} = \Delta L^g_{con,t}/I, \Delta L^{e,g}_{con,t} = I\Delta L^e_{con,t} \\ \mu^{e,g}_{con,t} + \mu^{g,e}_{con,t} \leq 1 \end{cases} \tag{4}$$

In the above formula: $L^{e,n}_{con,t}$ and $L^{g,n}_{con,t}$ are the power before the convertible electrical load and convertible natural gas load response; $L^e_{con,t}$ and $L^g_{con,t}$ are the power after the response of the convertible electrical load and convertible natural gas load; $\mu^{e,g}_{con,t}$ and $\mu^{g,e}_{con,t}$ are 0–1 auxiliary variables for interactive response; $\Delta L^e_{con,t}$ and $\Delta L^g_{con,t}$ are convertible electrical load and convertible natural gas load response quantities; $\Delta L^{g,e}_{con,t}$ and $\Delta L^{e,g}_{con,t}$ are the increased power of the convertible electrical load and convertible natural gas load after the response; $I$ is the electricity–gas conversion coefficient, which is taken as 1.25 in this paper.

### 2.2.2. Reducible Electrical Load Model

The reducible electrical load is the load that users can partially reduce [7]. The model is as follows:

$$\begin{cases} L^e_{adj,t} = L^{e,n}_{adj,t} - \mu^e_{adj,t}\Delta L^e_{adj,t} \\ 0 \leq \Delta L^e_{adj,t} \leq L^{e,n}_{adj,t} \end{cases} \tag{5}$$

In the above formula: $L^{e,n}_{adj,t}$ is the load power that can be reduced before the users respond; $\mu^e_{adj,t}$ is a 0–1 variable of whether to reduce; $\Delta L^e_{adj,t}$ is the load power actually reduced by the users.

Considering that power load reduction has a great impact on user satisfaction, the maximum duration of power load reduction is constrained in this paper:

$$\sum_{\tau=t}^{t+t^e_{adj,max}} (1 - \mu^e_{adj,\tau}) \geq 1 \; t = 1, 2, \cdots, T - t^e_{adj,max} \tag{6}$$

In the above formula: $t^e_{adj,max}$ is the maximum duration of electrical load reduction.

### 2.2.3. Transferable Electrical Load Model

After the community publishes the electricity price, users will transfer part of the electrical load from higher to lower hours to reduce the cost of energy use, such as washing

machines, electric vehicles, and other loads. This part of the load is called a transferable power load [7]. The transferable electrical load model is as follows:

$$
\begin{cases}
L_{\text{tran},t}^{\text{e}} = L_{\text{tran},t}^{\text{e,n}} + L_{\text{tran},t}^{\text{e,in}} - L_{\text{tran},t}^{\text{e,out}} \\
\sum_{t=1}^{T} L_{\text{tran},t}^{\text{e,in}} = \sum_{t=1}^{T} L_{\text{tran},t}^{\text{e,out}} \\
0 \leq L_{\text{tran},t}^{\text{e,in}} \leq L_{\text{tran,max}}^{\text{e,in}} \\
0 \leq L_{\text{tran},t}^{\text{e,out}} \leq L_{\text{tran,max}}^{\text{e,out}}
\end{cases}
\tag{7}
$$

In the above formula: $L_{\text{tran},t}^{\text{e,n}}$ and $L_{\text{tran},t}^{\text{e}}$ are the power before and after the response of the transferable electrical load at time $t$; $L_{\text{tran},t}^{\text{e,in}}$ and $L_{\text{tran},t}^{\text{e,out}}$ are the actual transfer-in and transfer-out power of the transferable electrical load at time $t$; $L_{\text{tran,max}}^{\text{e,in}}$ and $L_{\text{tran,max}}^{\text{e,out}}$ are the maximum load that can be transferred in and out at time $t$.

### 2.2.4. Heat Load Model

Some users in the community have high requirements for thermal comfort, the indoor temperature cannot be adjusted, and they are willing to bear the additional cost of thermal comfort. This part of the heat load is a fixed heat load. Another part of the user is willing to adjust the thermal comfort range, which is an adjustable heat load.

The adjustable heat load adopts the first-order building thermodynamic model [26]:

$$
L_{\text{adj},t}^{\text{h}} = N_1 \frac{1}{R} \left( \frac{T_{\text{in},t+1} - e^{-\Delta t/\tau_1} T_{\text{in},t}}{1 - e^{-\Delta t/\tau_1}} - T_{\text{out},t} \right)
\tag{8}
$$

In the above formula: $N_1$ is the number of users with adjustable heating temperature; $\tau_1 = R C_{\text{air}}$, $C_{\text{air}}$ is the heat capacity of indoor air, which can be taken as 1.2 kWh/$^\circ$C, $R$ is the equivalent thermal resistance of the house, which can be taken as 6.8 $^\circ$C/kW; $T_{\text{in},t}$ is the indoor temperature of the heating that can be adjusted at time $t$; $T_{\text{out},t}$ is the outdoor temperature.

The interactive response potential of heat load is mainly related to the human body's heat-using psychology for temperature perception, which has a certain elasticity. To better describe the user's thermal response potential, this paper introduces predicted mean vote (PMV), and the relationship between room temperature and PMV index value $\lambda_{\text{PMV},t}$ is as follows [27]:

$$
T_{\text{in},t} = T_{\text{com,s}} - \frac{M(2.43 - \lambda_{\text{PMV},t})(\lambda_{\text{clo}} + 0.1)}{3.76}
\tag{9}
$$

In the above formula: $T_{\text{com,s}}$ is the average temperature of human skin in a comfortable state, which can be taken as 33.5 $^\circ$C; $\lambda_{\text{clo}}$ and $M$ are the thermal resistance of the clothes and the metabolic rate of the human body, take 0.11 (m$^2 \cdot ^\circ$C)/W and 80 W/m$^2$ respectively.

Considering the recommendations of the ISO-7730 standard and the daily routine of the users, this paper limits the time sharing of the PMV index value, which is expressed as:

$$
\begin{cases}
|\lambda_{\text{PMV},t}| \leq 1, t \in [1,7] \cup [21,24] \\
|\lambda_{\text{PMV},t}| \leq 0.5, t \in [8,20]
\end{cases}
\tag{10}
$$

The fixed heat load model is:

$$
L_{\text{fir},t}^{\text{h}} = N_2 \frac{1}{R} \left( \frac{T_{\text{set}} - e^{-\Delta t/\tau_1} T_{\text{set}}}{1 - e^{-\Delta t/\tau_1}} - T_{\text{out},t} \right)
\tag{11}
$$

In the above formula: $N_2$ is the number of users with non-adjustable heating temperature; $T_{\text{set}}$ is the most comfortable indoor temperature of the users who cannot be adjusted.

To prevent the indoor temperature of adjustable users from being lower than the most comfortable temperature, the following constraints are imposed on the indoor temperature:

$$\sum_{t=1}^{T} \frac{T_{\text{in},t}}{T} = T_{\text{set}} \tag{12}$$

To sum up, the refined model of interaction considering user-side energy conversion is described as a matrix as follows:

$$\begin{bmatrix} L_t^{\text{e}} \\ L_t^{\text{h}} \\ L_t^{\text{g}} \end{bmatrix} = \begin{bmatrix} P_{\text{out},t}^{\text{e}} \\ P_{\text{out},t}^{\text{h}} \\ P_{\text{out},t}^{\text{g}} \end{bmatrix} = \begin{bmatrix} L_{\text{fir},t}^{\text{e}} \\ L_{\text{fir},t}^{\text{h}} \\ L_{\text{fir},t}^{\text{g}} \end{bmatrix} + \begin{bmatrix} L_{\text{adj},t}^{\text{e}} \\ L_{\text{adj},t}^{\text{h}} \\ 0 \end{bmatrix} + \begin{bmatrix} L_{\text{tran},t}^{\text{e}} \\ 0 \\ 0 \end{bmatrix} + \begin{bmatrix} L_{\text{con},t}^{\text{e}} \\ 0 \\ L_{\text{con},t}^{\text{g}} \end{bmatrix} \tag{13}$$

In the above formula: $L_t^{\text{e}}$, $L_t^{\text{h}}$ and $L_t^{\text{g}}$ are the electricity load, heat load, and natural gas load of the users; $L_{\text{fir},t}^{\text{e}}$ and $L_{\text{fir},t}^{\text{g}}$ are fixed electrical load and natural gas load; $L_{\text{fir},t}^{\text{h}}$ is the fixed heat load.

## 3. Optimization Model of Multi-Energy Transaction Decision between the Community Operator and the Users

### 3.1. Model Architecture of the Multi-Energy Transaction between the Community Operator and the Users Based on the Master–Slave Game

The master–slave game is an effective method to solve the problem of how to make decisions when there is an interest relation or conflict. The master–slave game is a dynamic non-cooperative game, and the unequal status of participants is the most fundamental difference between the master–slave game and the classical game. In the master–slave game, each subject has a different status and decision-making sequence. The leader has a leadership advantage and can occupy the first or advantageous position in the game, and the follower must follow the leader to make decisions. Not only does the retail price of energy set by the community operator affect consumer demand, but demand also affects price. Both parties have independent interest demands, and both make decisions with the goal of maximizing their own interests and influencing each other, and there is a master–slave game relationship.

The interaction relationship between the community operator and the users based on the master–slave game is shown in Figure 2. In the energy market environment, the community operator guides users to interactively use energy by setting reasonable retail energy prices. The energy purchase strategies of the energy market and equipment output are optimized under the maximum profit, and the users adjust the interactive energy use strategies according to the energy sales price of the community operator. In the master–slave game, the community operator, as the leader, guides users to interactively use energy by adjusting the retail prices of electricity and heat energy. As followers, users who receive retail prices of electricity and heat energy released by the community operator will change their energy usage habits to a certain extent and adjust their interactive energy usage strategies. At the same time, the users' changes in their own energy use needs will affect the retail energy price formulation strategies of the community operator. Additionally, the cycle goes back and forth when neither the community operator nor the users can improve their own interests by changing their own strategies; the equilibrium solution of the master–slave game model is reached. Finally, the community operator determines the final retail energy prices, the energy purchased in the energy market, and the output of each device. The users determine the interactive energy use strategies according to the retail energy prices released by the community operator.

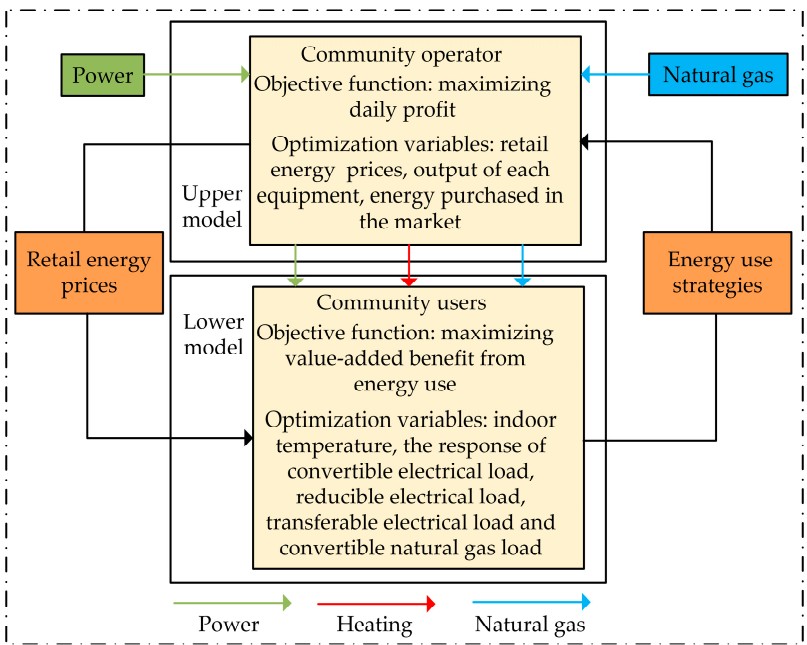

**Figure 2.** Multi-energy transaction decision-making model architecture between the community operator and the users.

### 3.2. Pricing Model of Community Operator's Retail Energy Prices

3.2.1. Objective Function

The community operator guides users to interact, optimize energy transactions, and maximize daily profit by formulating reasonable retail energy prices in the upper-level model. The objective function is:

$$C_1 = \max(C_{\text{sell}} + C_{\text{add}} - C_{\text{ob}} - C_{\text{mend}} - C_{\text{wv}}) \tag{14}$$

In the above formula: $C_1$ is the daily profit of the community operator; $C_{\text{sell}}$ is the energy retail income; $C_{\text{add}}$ is the additional income for thermal comfort; $C_{\text{ob}}$ is the cost of purchasing energy; $C_{\text{mend}}$ is the cost of equipment operation and maintenance; $C_{\text{wv}}$ is the cost of renewable power abandonment.

(1) Energy retail income

$$C_{\text{sell}} = \sum_{t=1}^{T} \sum_{i \in E} F_{i,t} L_t^i \tag{15}$$

In the above formula: $F_{i,t}$ is retail energy prices of the community; $E$ is the load type of the users, $E = \{e, h, g\}$.

(2) Additional income for thermal comfort

$$C_{\text{add}} = \sum_{t=1}^{T} F_{\text{add}} L_{\text{fir},t}^{\text{h}} \tag{16}$$

In the above formula: $F_{\text{add}}$ is the additional unit service price at which the heating load is not adjustable for the users to maintain the most comfortable temperature.

(3) Cost of purchasing energy

$$C_{\text{ob}} = \sum_{t=1}^{T} [F_{\text{net},e,t} P_{\text{net},t}^{\text{e}} + F_{\text{net},g,t} P_{\text{in},t}^{\text{g}}] \tag{17}$$

In the above formula: $F_{\text{net},e,t}$ and $F_{\text{net},g,t}$ are the unit cost of purchasing electricity and natural gas at time $t$.

(4)　Cost of equipment operation and maintenance

$$C_{\text{mend}} = \sum_{t=1}^{T} \sum_{j=1}^{J} F_{\text{m},j} P_{j,t} \tag{18}$$

In the above formula: $F_{\text{m},j}$ is the unit operation and maintenance costs of equipment $j$; $P_{j,t}$ is the output of equipment $j$ at time $t$.

(5)　Cost of renewable power abandonment

$$C_{\text{wv}} = \sum_{t=1}^{T} [\Delta P_{\text{w},t} F_{\text{w}} + \Delta P_{\text{v},t} F_{\text{v}}] \tag{19}$$

In the above formula: $\Delta P_{\text{w},t}$ and $\Delta P_{\text{v},t}$ are the amount of abandoned wind and light at time $t$; $F_{\text{w}}$ and $F_{\text{v}}$ are the unit cost of abandoned wind and light.

### 3.2.2. Multiple Energy Price Constraints

To prevent the community operator from maliciously raising prices and ensuring the benefit of users, multiple energy prices need to be constrained:

$$F_{i,t,\text{min}} \leq F_{i,t} \leq F_{i,t,\text{max}} \tag{20}$$

$$\sum_{t=1}^{T} \frac{F_{i,t}}{T} \leq F_{i,\text{av}} \tag{21}$$

In the above formula: $F_{i,t,\text{max}}$ and $F_{i,t,\text{min}}$ are the maximum and minimum prices for electricity and heat prices to protect the benefits of the users and the operator; $F_{i,\text{av}}$ is the average price of electricity and heat in the energy market.

### 3.2.3. Energy Conversion Equipment Constraints

The traditional energy conversion equipment model considers the output efficiency of the equipment to be a fixed value. To ensure the accuracy of the equipment output, this paper adopts the dynamic energy-efficiency model of energy conversion equipment. The output efficiency of energy conversion equipment is mainly related to the load rate [28,29]. It is expressed by means of polynomial fitting, as shown in Formula (22).

$$\eta_{x,n} = \sum_{k=0}^{n} \alpha_{x,k} \left( \frac{P_x}{P_{x,\text{N}}} \right)^k \tag{22}$$

In the above formula: $\eta_{x,n}$ is the dynamic energy efficiency of equipment $x$ using n-order polynomial fitting; $\alpha_{x,k}$ is the fitting coefficient; $P_x$ and $P_{x,\text{N}}$ are the actual output power and rated output power of the equipment.

The input–output relationship of energy conversion equipment considering dynamic energy efficiency is as follows:

(1)　Combined heat and power unit

The output power efficiency of the CHP unit can be fitted by a fourth-order fitting [28]:

$$P_{\text{CHP},t}^{\text{e}} = \eta_{\text{CHP},4} P_{\text{CHP},t}^{\text{g}} \tag{23}$$

In the above formula: $P_{\text{CHP},t}^{\text{g}}$ is the natural gas power entering the CHP unit; $P_{\text{CHP},t}^{\text{e}}$ is the output electric power of the CHP unit.

In this paper, the strategy of determining the heat by electricity is adopted, and the thermoelectric ratio $\psi_{\text{CHP}}$ can be described by the second-order fitting of the electrical load rate of the CHP unit:

$$\psi_{\text{CHP}} = \sum_{k=0}^{2} \left( \alpha_{\psi,k} N_{\text{CHP,e}}^{k} \right) \tag{24}$$

In the above formula: $N_{\text{CHP,e}}$ is the electrical load rate of the CHP unit.

Therefore, the output thermal power $P_{\text{CHP},t}^{\text{h}}$ of the CHP unit is:

$$P_{\text{CHP},t}^{\text{h}} = \psi_{\text{CHP}} P_{\text{CHP},t}^{\text{e}} \tag{25}$$

(2) Gas boiler

The output thermal efficiency of the gas boiler can be fitted by the first-order:

$$P_{\text{GB},t}^{\text{h}} = \eta_{\text{GB},1} P_{\text{GB},t}^{\text{g}} \tag{26}$$

In the above formula: $P_{\text{GB},t}^{\text{g}}$ is the natural gas power entering the gas boiler; $P_{\text{GB},t}^{\text{h}}$ is the output thermal power of the gas boiler.

(3) Electric heat pump

The output thermal efficiency of the electric heat pump is related to the load rate and temperature. In this paper, only the influence of the load rate is considered, and second-order fitting can be used [29]:

$$P_{\text{EHP},t}^{\text{h}} = \eta_{\text{EHP},2} P_{\text{EHP},t}^{\text{e}} \tag{27}$$

In the above formula: $P_{\text{EHP},t}^{\text{e}}$ is the electric power entering the electric heat pump; $P_{\text{EHP},t}^{\text{h}}$ is the output heat power of the electric heat pump.

### 3.2.4. Device Operation Constraints

(1) Energy Conversion Equipment Constraints

$$\begin{cases} P_i^{\min} \leq P_{i,t} \leq P_i^{\max} \\ -P_i^{\text{down}} \leq P_{i,t} - P_{i,t-1} \leq P_i^{\text{up}} \end{cases} \tag{28}$$

In the above formula: $P_i^{\max}$ and $P_i^{\min}$ are the upper and lower limits of the output of the coupling device at time $t$; $P_i^{\text{up}}$ and $P_i^{\text{down}}$ are the upper and lower limits of the climbing power of the coupling device at time $t$.

(2) Energy Storage Device Constraints

$$\begin{cases} S_{i,t}^{\text{ES}} = (1 - \sigma_i^{\text{ES}}) S_{i,t-1}^{\text{ES}} + P_{\text{c/d},i,t}^{\text{ES,n}} \Delta t \\ P_{\text{c/d},i,t}^{\text{ES,n}} = \eta_{\text{c},i}^{\text{ES}} P_{\text{c},i,t}^{\text{ES}} - \frac{1}{\eta_{\text{d},i}^{\text{ES}}} P_{\text{d},i,t}^{\text{ES}} \\ C_{i,\min}^{\text{ES}} \leq S_{i,t}^{\text{ES}} \leq C_{i,\max}^{\text{ES}} \\ 0 \leq P_{\text{c},i,t}^{\text{ES}} \leq \varepsilon_{\text{c},i,t} P_{\text{c},i,t}^{\max} \\ 0 \leq P_{\text{d},i,t}^{\text{ES}} \leq \varepsilon_{\text{d},i,t} P_{\text{d},i,t}^{\max} \\ S_{i,0}^{\text{ES}} = S_{i,T}^{\text{ES}} \end{cases} \tag{29}$$

In the above formula: $S_{i,t}^{\text{ES}}$ is the energy storage value of the energy storage device at time $t$; $\sigma_i^{\text{ES}}$ is the self-loss rate of the energy storage device; $\eta_{\text{c},i}^{\text{ES}}$ and $\eta_{\text{d},i}^{\text{ES}}$ are the charging and discharging efficiency of the energy storage device; $P_{\text{c},i,t}^{\text{ES}}$ and $P_{\text{d},i,t}^{\text{ES}}$ are the charging and discharging power of the energy storage device; $C_{i,\max}^{\text{ES}}$ and $C_{i,\min}^{\text{ES}}$ are the upper and lower limits of the capacity of the energy storage device; $\varepsilon_{\text{c},i,t}$ and $\varepsilon_{\text{d},i,t}$ are 0–1 auxiliary variables; $P_{\text{c},i,t}^{\max}$ and $P_{\text{d},i,t}^{\max}$ are the upper limit of energy storage charging and discharging power respectively.

### 3.2.5. Renewable Energy Output Constraints

In the below formula: $P_{w,t}^{e,pre}$ and $P_{v,t}^{e,pre}$ are the predicted power values of wind power and photovoltaic.

$$\begin{cases} 0 \le P_{w,t}^{e} \le P_{w,t}^{e,pre} \\ 0 \le P_{v,t}^{e} \le P_{v,t}^{e,pre} \end{cases} \tag{30}$$

In addition, the upper-layer model also needs to satisfy the power balance constraint of Equation (13). The above nonlinear model can be approximated by piecewise linearization.

### 3.3. Energy Use Strategies Model Considering User Interaction

According to the retail energy prices released by the community operator, users consider increasing or decreasing load, converting load, and transferring load. For details, see the refined model of user interaction considering energy conversion in Section 2. The users' goal is to optimize their own interactive energy-use strategies and maximize the value-added benefit of energy use. This paper defines the value-added benefit of a users' energy use, which is the users' total energy use utility $C_{ben}$ minus the total cost. The total cost includes the cost of purchasing energy $C_{ub}$ and the additional cost of thermal comfort $C_{app}$.

$$C_2 = \max[C_{ben} - (C_{ub} + C_{app})] \tag{31}$$

$$C_{ben} = \sum_{t=1}^{T} \sum_{i \in E} [f_1^i L_t^i - \frac{f_2^i}{2}(L_t^i)^2] \tag{32}$$

$$C_{ub} = C_{sell} = \sum_{t=1}^{T} \sum_{i \in E} F_{i,t} L_t^i \tag{33}$$

$$C_{app} = C_{add} = \sum_{t=1}^{T} F_{add} L_{fir,t}^{h} \tag{34}$$

In the above formula: $f_1^i$ and $f_2^i$ are the constant coefficients of the users' preference for type $i$ energy, reflecting the users' preference for energy demand.

### 3.4. Model-Solving Process

According to the reference [30], it is easy to prove that there is a unique equilibrium solution for the optimization model of multi-energy transaction decisions between the community operator and the users based on the master–slave game in this paper.

This paper uses MATLAB software programming, and the method of combining the differential evolution algorithm and CPLEX solver is used to solve the proposed model. The solution process is as follows:

(1) Initialize the parameters of the community operator and the users, $k = 0$, set the maximum number of iterations $k_{max} = 100$, use the differential evolution algorithm to randomly generate retail energy prices of 10 groups of the community operator, and transmit them to the energy use strategies model considering user interaction.

(2) $k = k + 1$.

(3) The users receive retail energy prices published by the community operator. Use the CPLEX solver to solve the energy use strategies model and the optimal value-added benefit $C_2^k$, and return the energy use strategies to the model of the community operator.

(4) The community operator optimizes the output of equipment and the amount of electricity and gas purchased in the market according to the energy use strategies of the users and calculates the optimal profit $C_1^k$ of the community operator.

(5) Use the variation and crossover of the differential evolution algorithm to generate a group of new retail energy prices and repeat the processes in (3) and (4). Additionally, calculate the optimal value-added benefit $C_2^{k*}$ of the users and the optimal profit $C_1^{k*}$ of the community operator under the new retail energy prices.

(6) Perform selection operation: compare the optimal solutions of the community operator before and after mutation and crossover; if $C_1^{k*} \geq C_1^k$, then $C_1^{k+1} = C_1^{k*}$, $C_2^{k+1} = C_2^{k*}$; if $C_1^{k*} < C_1^k$, then $C_1^{k+1} = C_1^k$, $C_2^{k+1} = C_2^k$.

(6) If $k \geq k_{\max}$, end the program; otherwise, return to flow (2).

## 4. Case Analysis

### 4.1. Parameter Settings

This paper selects a community integrated energy system in the northern winter of China as the research object. The forecasting curves of the renewable energy output, initial load, and outdoor temperature curves are shown in Figure 3. Other required parameter data are shown in the table in the Appendix A. The equipment parameters of the community operator are shown in Table A1, which is used to deal with the community operator's device model. The data parameters for the users are shown in Table A2, which is the parameter data required by the user model. The equipment efficiency fitted from Table A2 is shown in Figure A1. The time-of-use electricity price in the energy market is shown in Table A3 [31], it is the price that the community operator trades with the energy market. To protect the benefit of the users, the upper limit of the electricity price set by the community operator is not higher than the time-of-use electricity price in the energy market; the lower limit is not less than 0.2 RMB/(kWh). The heat price of the energy market is 0.35 RMB/(kWh) [32], and the thermal price range set by the community ranges from 0.2 to 0.5 RMB/(kWh). The natural gas price within the community is the same as the market gas price, which is 0.34 RMB/(kWh) [33].

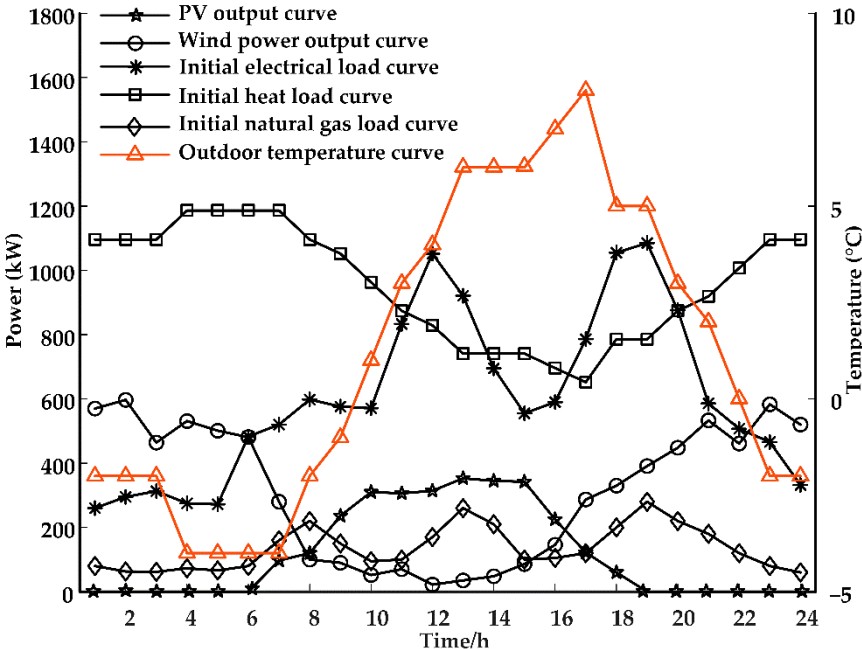

**Figure 3.** Forecasting curves of renewable energy output, initial load, and outdoor temperature.

### 4.2. Scene Settings

To illustrate that the multi-energy trading strategies proposed in this paper can improve the profit of the community operator and the value-added benefit of the users, and can improve renewable energy utilization, three scenarios are set for comparative analysis:

- Scenario 1: Regardless of user interaction [6], the energy price sold by the community operator to the users is the market price.
- Scenario 2: Considering user interaction [15] and ignoring the influence of the users' electricity–gas convertible load, the community operator and the users compete to determine the electricity price and heat price of the community.

- Scenario 3: Considering user interaction, using the refined model of user interaction considering energy conversion and considering the impact of electricity–gas convertible load, the community operator and the users play games to formulate the internal electricity price and heat price in the community.

*4.3. Simulation Analysis*

4.3.1. Analysis of the Multi-Energy Transaction Results of the Community Operator

Under the model proposed in this paper in Scenario 3, the electricity and heat prices traded between the community operator and the users are shown in Figure 4.

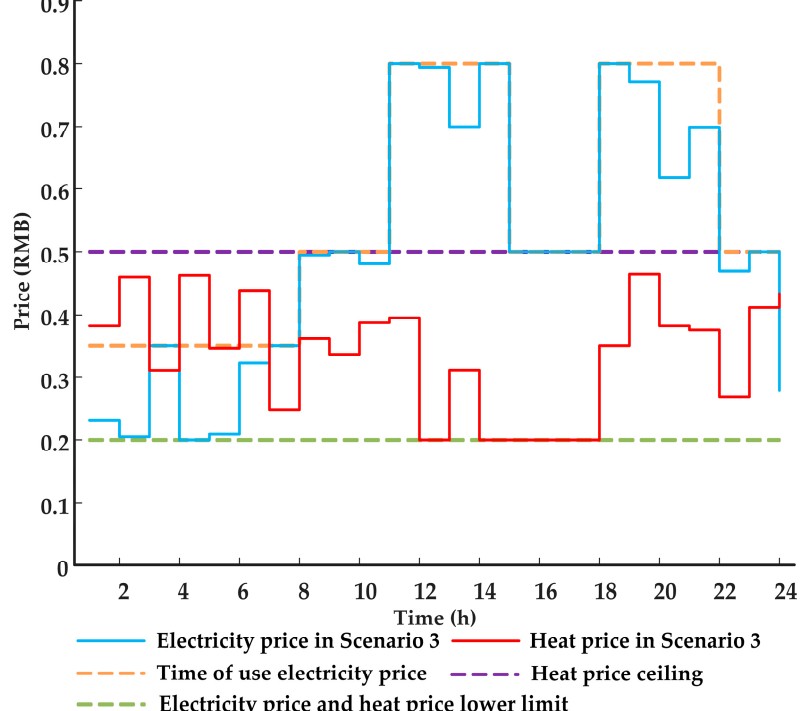

**Figure 4.** The price of electricity and heat that the operator trades with users.

The electricity price trend set by the community operator basically follows the electricity price in the energy market, and the peak electricity price is in the peak energy use period, which is in line with the actual situation. The heat price set by the community operator fluctuates within the limit of the heat price. Judging from the electricity and heat prices traded by the community operator and the users in Figure 4, the average price of electricity set by the community operator in Scenario 3 is 0.5 RMB/(kWh), the average price of electricity in the energy market is 0.56 RMB/(kWh). The average price of heat within the community is 0.33 RMB/(kWh), and the average price of heat in the energy market is 0.35 RMB/(kWh). Based on the above analysis, the average electricity price and average heat price set by the community operator are 10.7% and 5.7% lower, respectively, than in the market. In Figure 4, in Scenario 3, the electricity and heat prices traded between the community operator and the users can protect the benefit of the users and promote energy transactions between the users and the community operator.

Figure 5 shows the results of electricity and gas transactions between the community operator and the energy market in different scenarios. Taking the peak period of electricity price as an example, the following analysis is made: from Figure 5, it can be seen that in Scenarios 2 and 3, compared with Scenario 1, the community operator purchases less electricity from the power grid during the peak period of electricity price, which effectively reduces the power supply pressure on the large power grid, indicating that user interaction can indirectly participate in the power market and reduce the peak power consumption of the large power grid. Compared with Scenario 2, Scenario 3 considers the convertible

load, and part of the electricity load demand of the users is supplied by natural gas during the peak electricity price period. The energy market has the least amount of electricity purchased and the largest amount of gas during the corresponding period.

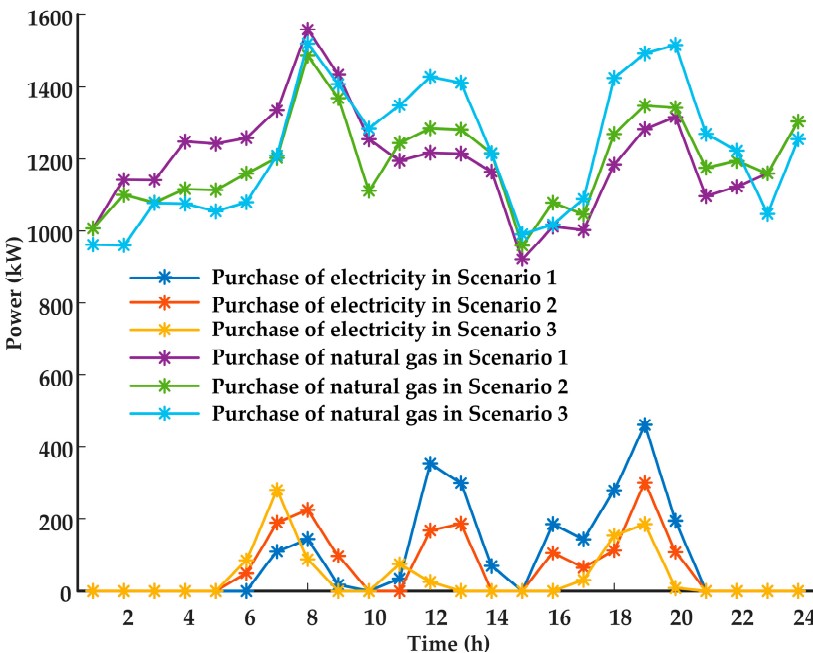

**Figure 5.** Electricity and natural gas trading results between the community operator and the energy market in different scenarios.

### 4.3.2. Analysis of Energy Use Strategies for User Interaction

Figure 6 shows the results of electricity and natural gas transactions between the community operator and the users in different scenarios. Scenario 1 does not consider user interaction, and its electricity and natural gas loads are all original loads. As shown in Figure 6, in both Scenarios 2 and 3, considering user interaction, the electrical load peak smoothed to varying degrees. Scenario 1 has the highest electrical load peak, followed by Scenario 2. Scenario 3 adopts the strategies of this paper, with the smallest electrical load peak and the largest gas load fluctuation.

Considering the limitations of space, this paper will focus on the analysis of the interactive energy use strategies of users' electricity–gas convertible load. In Scenario 3, the amount of the convertible load interaction, including consumer psychology, is shown in Figure 7, which shows the relationship between the users' actual convertible load response and the equivalent energy price difference. This is consistent with the optimized results in Figure 6. For users, when the electricity price is lower than the equivalent natural gas price (the natural gas price multiplied by the electricity–gas conversion coefficient). For example, to meet the same demand for hot water, the electricity cost of the users is lower than the natural gas cost. When the natural gas price is high, the cost of electricity is higher than the cost of natural gas. Therefore, when the electricity price is lower than the equivalent natural gas price, users replace part of the convertible natural gas load with electricity, and when the electricity price is higher than the equivalent natural gas price, users replace part of the convertible electricity load with natural gas. Compared with Scenario 2, the convertible load is considered in Scenario 3, and the electricity load curve during the peak period of the electricity price during the periods of 11:00~14:00 and 18:00~21:00 is lower than that of Scenario 2. In the low electricity price period from 1:00 to 7:00, the power load curve of Scenario 3 is higher than that of Scenario 2. After considering the convertible load, the electrical load curve of the users in Scenario 3 is smoother, and the outline of the electric load curve is optimized.

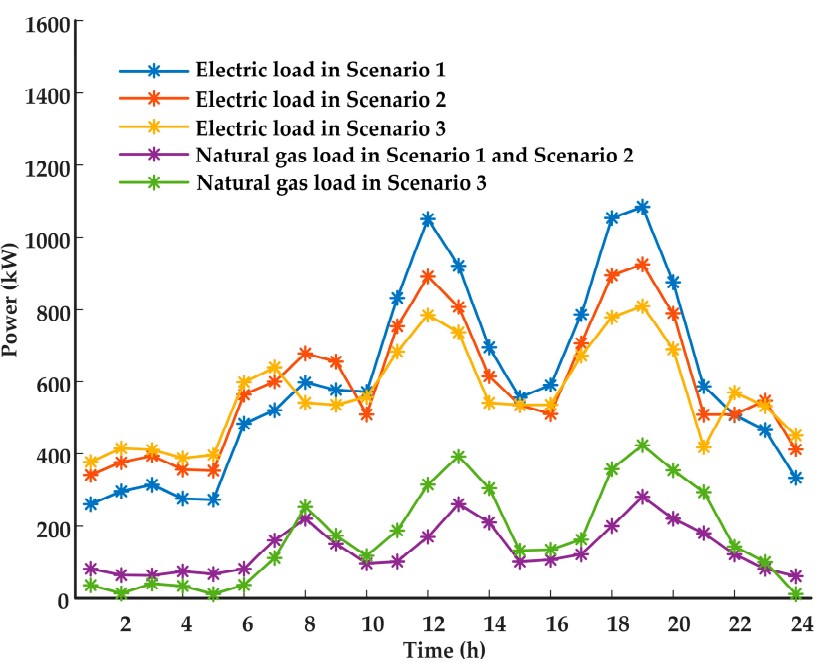

**Figure 6.** Electricity and natural gas loads of users in different scenarios.

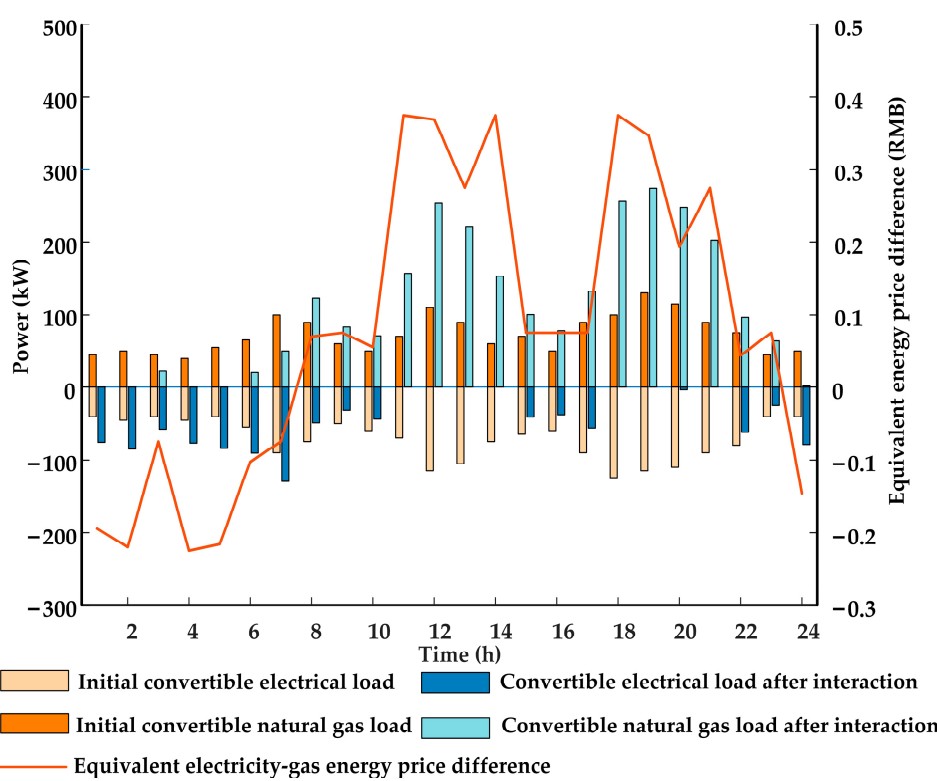

**Figure 7.** The relation between the convertible load and the equivalent energy price difference.

When the users' heat load does not participate in the interaction, that is, the original heat load (Scenario 1), the indoor temperature is kept at the most comfortable temperature. When considering the users' adjustable heat load, the indoor temperature is kept in a suitable range (Scenario 3 is used as an example). The heat load and indoor temperature of Scenarios 1 and 3 are shown in Figure 8.

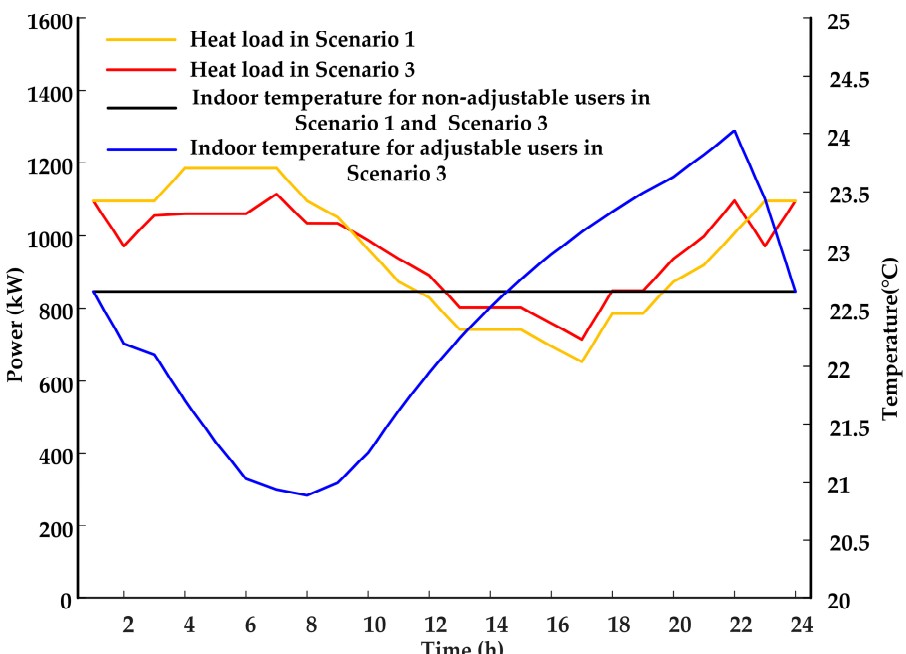

**Figure 8.** Heat load and indoor temperature in Scenarios 1 and 3.

According to Figure 8, the indoor temperature of users in Scenarios 1 and 3 cannot be adjusted at room temperature and is always maintained at about 22.6 degrees Celsius. The indoor temperature of users in Scenario 3 can be adjusted to maintain the indoor temperature within 20.5 to 24.5 degrees Celsius, ensuring the users' thermal comfort. From time period 1 to time period 9, the heat load provided by Scenario 3 is lower than the original heat load, and coupled with heat loss, the temperature decreases. At the same time, to keep the temperature of the first and last sections consistent, the heat load adjustment in the last few periods is relatively large.

### 4.3.3. Cost-Benefit Analysis of the Community Operator and Users

To show the superiority of the model and trading strategy proposed in this paper, the profit and profit rate of the community operator, the cost of the users, and the value-added benefit are compared in different scenarios. The comparison results are shown in Table 1.

**Table 1.** Comparative Study 1.

| Compare Items | | Scenario 1 | Scenario 2 | Scenario 3 |
|---|---|---|---|---|
| Community operator | Cost (RMB) | 13,940 | 12,486 | 11,976 |
| | Profit (RMB) | 4459 | 4329 | 4720 |
| | Profit margin (%) | 31.9 | 34.7 | 39.4 |
| | Renewable energy utilization (%) | 88 | 93 | 96 |
| Community users | Total cost (RMB) | 18,399 | 16,815 | 16,690 |
| | Value-added benefit | 15,121 | 16,993 | 17,412 |

Overall, it can be seen from Table 1 that, compared with Scenarios 1 and 2, Scenario 3 is the establishment of the multi-energy prices within the community under the model proposed in this paper, the profit and profit margin of the operator, the value-added benefit of the users' energy use, and the renewable energy utilization have increased significantly.

The specific analysis is as follows: In terms of on-site consumption of renewable energy, the on-site renewable energy utilization in Scenario 2 and Scenario 3 has increased by more than 5% compared with Scenario 1. Scenario 3 has the highest renewable energy utilization, followed by Scenario 2 and Scenario 1. Therefore, user interaction can improve

on-site renewable energy utilization. In terms of the community operator's profit and the energy value-added benefit of the users, compared with Scenario 1, Scenario 2 reduces the community operator's profit by 130 RMB. The cost of the users' energy use has been reduced by 1584 RMB, and the value-added benefit of the users' energy use has been greatly improved. This is because Scenario 1 is sold to the users at the market energy prices, the retail energy prices are relatively high, and the users are not given preferential energy prices. Scenario 2 considers user interaction and gaming, the retail price of energy in the community is determined, and the energy prices are constrained within an appropriate range, which can greatly improve user satisfaction with energy use. Compared with Scenario 2, the community operator's profit in Scenario 3 has increased by 391 RMB. After converting the load, the operator reduces the penalty cost of renewable energy curtailment and the power supply cost during peak power consumption. For the users, the total cost is reduced by 125 RMB after considering the convertible load. At the same time, the value-added benefit of the users' energy use in Scenario 3 compared with Scenario 2 has been improved, and the satisfaction with energy use has been further improved. Compared with Scenarios 1 and 2, Scenario 3 has the highest profit for the community operator, the lowest total energy use cost for the users, and the highest value-added benefit. Although the community operator in Scenario 3 has the lowest turnover, it has the lowest cost. From the profit side, the profit in Scenario 3 is 5.9% and 9% higher than that in Scenario 1 and Scenario 2, respectively. In terms of profit margin, Scenario 3 has the highest profit margin, which is 7.5% and 4.7% higher than Scenario 1 and Scenario 2, respectively. Meanwhile, the value-added benefit of the users in Scenario 3 is 15.2% and 2.5% higher than that in Scenario 1 and Scenario 2, respectively. Scenario 3 can maximize the benefits of the community operator and the users. The community operator can actively guide user interaction during pricing by optimizing retail energy prices, improving the users' value-added benefit of energy use, and at the same time increasing the community operator's own profit and making the users more satisfied with the services provided by the community operator to consolidate and expand the user base and provide other value-added services. To sum up, Scenario 3 can consider the benefits of the community operator and the users and achieve a win–win situation.

To further clarify the work of this paper, Scenario 4 is added: a centralized optimization method is adopted to optimize the maximum profit of a community operator with a single objective [9], considering user interaction, but not considering the convertible load. The comparison results are shown in Table 2.

**Table 2.** Comparative Study 2.

| Compare Items | | Scenario 3 | Scenario 4 |
|---|---|---|---|
| Community operator | Cost (RMB) | 11,976 | 13,153 |
| | Profit (RMB) | 4720 | 4957 |
| | Profit margin (%) | 39.4 | 37.6 |
| Community users | Total cost (RMB) | 16,690 | - |
| | Value-added benefit | 17,412 | - |
| The optimization method | | The master–slave game | The centralized optimization |
| Whether user interaction is considered | | √ | √ |
| Whether the convertible load is considered | | √ | × |
| Whether the retail energy prices have been optimized | | √ | × |
| Whether a win–win situation has been achieved | | √ | × |

As can be seen from Table 2, Scenario 4 adopts the centralized optimization with the single goal of maximizing the profit of the community operator, without considering the interests of the users. From the optimization results, the profit of the community operator in Scenario 4 is higher than that in Scenario 3 because the community operator does not offer preferential energy prices to the users and fails to balance the interests of users in Scenario 4.

Meanwhile, the cost of the community operator in Scenario 3 is lower than that in Scenario 4, and the profit margin of the community operator in Scenario 3 is 1.8% higher than that in Scenario 4. In Scenario 3, the community operator can optimize the energy prices and balance the value-added benefit of the users through appropriate profit sharing. Although the profit is reduced, it can improve the users' satisfaction with the use of energy and improve the profitability of the community operator. In terms of user interaction, Scenario 3 considers the convertible load, which can enrich the way users use energy. Under the optimization method of the master–slave game in Scenario 3, the community energy prices can be optimized, and the win–win situation between the community operator and the users can be realized.

Through the above comparative analysis, using the multi-energy transaction decision optimization model of the community operator and the users considering user interaction proposed in this paper, it is possible to formulate reasonable retail energy prices for the community, guide the users to interact, and improve the community operator's profit and value-added benefit of the users' energy use to achieve a win–win situation.

## 5. Conclusions

The master–slave game model constructed in this paper describes the energy transaction between the community operator and the users and proposes the optimization model of the multi-energy transaction decision between the community operator and the users. The upper model considers the maximum profit of the community operator, while the lower model aims for the maximum value-added benefit of the users. The model is effectively analyzed by an example, and the relevant conclusions are as follows:

(1) The retail energy prices of the community determined by the decision in this paper are reasonable and acceptable to the users. The average price of electricity and the average price of heat set by the community operator are 10.7% and 5.7% lower, respectively, than the market, which protects the interests of the users. The model is extensible. By modifying the corresponding model, more user groups can be promoted. For different countries and regions, the model of the upper community operator can be modified according to the actual situation, including the type of equipment and the type of renewable energy, which can be adjusted according to the needs. The lower user model can determine the types of user interaction load (including reducible, transferable, transferable, etc.) according to the living habits of residents in different countries and regions. At the same time, the lower model has variable data related to the users and can be applied to residential communities with different energy preferences.)

(2) With the continuous improvement of the user side equipment, convertible load becomes possible. Users can choose appropriate energy modes to meet their energy needs according to different energy prices. The refined user interaction model that considers energy conversion constructed in this paper can reduce user costs.

(3) The optimization model of the multi-energy transaction decision between the community operator and the users proposed in this paper considers the energy conversion on the user side, which can not only improve the profit of the community operator, but also increase the value-added benefit of energy use and realize a win–win situation for the community operator and the users. Using the strategy proposed in this paper to set the community prices increases the community operator's profit and profit margin by 5.9% and 7.5%, respectively, compared to using market energy prices directly. At the same time, the value-added benefit to users also increases by 15.2%. In addition, user interaction can indirectly reduce the peak value of the grid, which is beneficial to grid security.

The model established in this paper mainly formulates the retail prices of community energy from the dimensions of the community and the users, ignoring the energy connection between the community and the community. At the same time, the impact of load and renewable energy uncertainty is ignored. The next step will continue to study the

energy transaction strategies between the community and the community, and the impact of uncertainty. In terms of the community and the community, a single community may have an energy surplus or a shortage at some time. Multi-communities can trade surplus or shortage energy according to a certain mode to realize the efficient use of resources. Uncertainties also affect the decisions of the community operator. We will continue to study these two aspects in future research.

**Author Contributions:** Conceptualization and methodology, Y.L. and X.W.; simulation and analysis, Y.L.; investigation, X.W.; data curation, Y.L.; writing—original draft preparation, Y.L.; writing—review and editing, Y.L. and X.W.; supervision, X.W.; literature research, Y.L. All authors have read and agreed to the published version of the manuscript.

**Funding:** This research was supported by the Natural Science Foundation of Xinjiang Uygur Autonomous Region under Grant 2020D01C031.

**Institutional Review Board Statement:** Not applicable.

**Informed Consent Statement:** Not applicable.

**Data Availability Statement:** Not applicable.

**Conflicts of Interest:** The authors declare no conflict of interest.

## Appendix A

**Table A1.** The equipment parameters.

| Equipment | Parameter Type | Parameter Value |
|---|---|---|
| CHP | Rated Capacity | 300 kW |
| | Minimum output power | 100 kW |
| | Electrical efficiency fitting coefficient | $\alpha_{CHP,0} = 0.09$, $\alpha_{CHP,1} = 0.44$, $\alpha_{CHP,2} = -0.14$, $\alpha_{CHP,3} = -0.11$, $\alpha_{CHP,4} = 0.06$ |
| | Thermoelectric ratio fitting coefficient | $\alpha_{\psi,0} = 3.82$, $\alpha_{\psi,1} = -5.84$, $\alpha_{\psi,2} = 3.6$ |
| | Operation and maintenance cost | 0.04 RMB/(kWh) |
| EHP | Rated Capacity | 200 kW |
| | Thermal efficiency fitting coefficient | $\alpha_{EHP,0} = 2.61$, $\alpha_{EHP,1} = 0.36$, $\alpha_{EHP,2} = 0.026$ |
| | Operation and maintenance cost | 0.06 RMB/kWh |
| GB | Rated Capacity | 1000 kW |
| | Thermal efficiency fitting coefficient | $\alpha_{GB,0} = 0.81$, $\alpha_{GB,1} = 0.13$ |
| | Operation and maintenance cost | 0.02 RMB/kWh |
| Electricity storage | Rated Capacity | 500 kW |
| | Charge/Discharge efficiency | 0.98 |
| | Attrition rate | 0.02 |
| | Operation and maintenance cost | 0.01 RMB/kWh |
| Heat storage | Rated Capacity | 500 kW |
| | Charge/Discharge efficiency | 0.95 |
| | Attrition rate | 0.02 |
| | Operation and maintenance cost | 0.01 RMB/kWh |

**Table A2.** Data parameters for users.

| Parameter | Meaning | Value | Parameter | Meaning | Value |
|---|---|---|---|---|---|
| $\pi_{con,min}^{e,g}$ | Dead threshold for electrical energy conversion | 0 | $\pi_{con,max}^{e,g}$ | Saturation value of electrical energy conversion | 0.2 |
| $\pi_{con,min}^{g,e}$ | Dead threshold for natural gas conversion | 0 | $\pi_{con,max}^{g,e}$ | Saturation value of natural gas conversion | 0.15 |
| $f_1^e$ | First power coefficient of electrical energy preference | 1.5 | $f_2^e$ | Quadratic coefficient of electrical energy preference | 0.0009 |
| $f_1^h$ | First power coefficient of thermal energy preference | 1.1 | $f_2^h$ | Quadratic coefficient of thermal energy preference | 0.0011 |
| $f_1^g$ | First power coefficient of natural gas preference | 1.2 | $f_2^g$ | Quadratic coefficient of natural gas preference | 0.001 |
| $N_1$ | Number of the users with adjustable heating temperature | 500 | $N_2$ | Number of the users with non-adjustable heating temperature | 300 |
| $L_{tran,max}^{e,out}$ | Maximum load that can be transferred out | 80 kW | $L_{tran,max}^{e,in}$ | Maximum load that can be transferred in | 80 kW |
| $t_{adj,max}^e$ | Maximum duration of electrical load reduction | 4 h | $T_{set}$ | The most comfortable indoor temperature | 22.6 °C |

**Table A3.** Time of use electricity prices.

| Period | Market Electricity Price (RMB/kWh) |
|---|---|
| Valley period: 01:00—07:00 | 0.35 |
| Normal period: 08:00—10:00; 15:00—17:00; 22:00—24:00 | 0.5 |
| Peak period: 11:00—14:00; 18:00—21:00 | 0.8 |

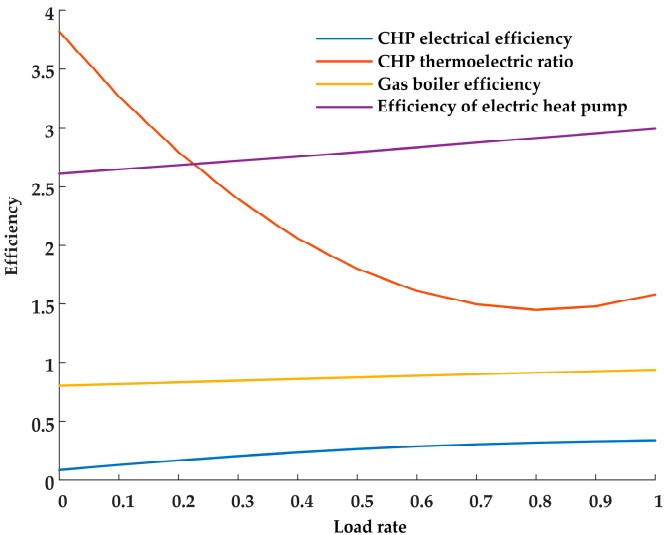

**Figure A1.** Fitting curve of equipment efficiency.

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
