# Peer review of "Community Integrated Energy System Multi-Energy Transaction Decision Considering User Interaction"

_processes, doi:10.3390/pr10091794_

Round 1
Reviewer 2 Report
Dear authors, attached you can find comments from my review. Best regards

Reviewer 3 Report
This paper is well written and has a good scope in terms of the integration of renewable energy resources. The paper introduces novel ideas including the operator problems and convertible load in the energy trading sector. Overall the paper has much scope in the energy sector, but the following improvements are suggested to improve the impact of paper.
Kindly, consider revising title which should be easily understood by novice people. It will increase the impact of paper.
Please revise first sentence of “Abstract”, It seem grammatically wrong. Instead provide a comprehensive statement on background in one or two sentences in a simple language describing the broad benefits. Reader may be not much attracted to read further for the current sentence.
Also revise sentence of “Abstract”, “Based on the energy retail prices set by the community operator, optimize the users’ energy use strategies.”
Kindly add more references in “Introduction” to elaborate the maximum relevant work on the topic. Specifically, add more relevant references on the operator problems described in this paper and convertible load.
Please add the section wise paper organization in the end of “Introduction” For instance… “Section 2 summarizes the models of Community Integrated Energy System and of User Participation in interaction” , and so on.
Kindly provide reference for Equations used in Model. It should be clearly mentioned what part of equation was taken from literature and what are modified.
In section conclusion kindly elaborate the statement “The next step will continue to study the energy transaction strategies between the community and the community.” Also include limitations of this study and suggest more comprehensive directions on future extension of this work.
Kindly, compare the significance of model results in the end of “Last section just before conclusion” with reference to the existing literature review which was done in “Introduction”.
Reviewer 4 Report
In this study, a model is proposed to improve the economy of community integrated energy systems for community operators and users.
1. Introduction: instead of using the word ‘reference’, it would be better to use the author’s name for a more appealing impression.
2. Please include a brief description of the master-slave game for the understanding of new academic researchers.
3. What are the energy conversion efficiencies of equipment taken for this study?
4. Section 4.1, Please provide references for the prices taken.
5. What is the economic implementation of this study? How would this study benefit economic practitioners in taking decisions?
6. Section 5, please include future directions and limitations of this study.
7. The abstract section must be improved by including results and conclusions. Currently, the abstract seems to be a description of this study.
Round 2
Reviewer 1 Report
Dear authors,
my main comments seem to be answered appropriately.
In my opinion, the title is still too complicated.
In point two of my first review I meant there are articles missing - grammatically - like "the", and not references. Therefore I suggest an external grammar check.
Author Response
Dear Reviewer:
Many thanks to the reviewers for taking the time to review our paper and thank you very much for your valuable advice. We have carefully revised the article as you suggested. Your suggestion is of great help to us. We hope you will be satisfied with the result of our revision.
Best wishes.
Point 1: In my opinion, the title is still too complicated.
Response 1: Thank you very much for this important comment. In order to better understand the title of the article, we changed the title to “Community Integrated Energy System Multi-Energy Transaction Decision Considering User Interaction”. Thanks again for your valuable advice.
Point 2: In point two of my first review I meant there are articles missing - grammatically - like "the", and not references. Therefore I suggest an external grammar check.
Response 2: Thank you very much for this important comment. We are very sorry for the inconvenience caused to the reviewers. According to your suggestion, we have carefully checked and revised the model, words, grammar and other contents of this paper. And we asked the relevant English teachers to check our article to better understand the content of the articles. Thanks again to the reviewer for this valuable suggestion.
Reviewer 2 Report
Dear authors,
please find attached my notes.
Best regards

Author Response
Dear Reviewer:
Please see the attachment.
Best wishes.

Round 3
Reviewer 2 Report
Dear Authors,
the question remains why you do not make a demarcation/comparison to this work (Optimal Dispatch of Community Integrated Energy System Considering Comprehensive User Satisfaction Method - Liu - IEEJ Transactions on Electrical and Electronic Engineering - Wiley Online Library) or my comments on e.g. Figure 1? Please see the document from last time.
Best regards
Author Response
Dear reviewer:
Please see the attachment.
Best regards
